# Comparison of Sin-QuEChERS Nano and d-SPE Methods for Pesticide Multi-Residues in Lettuce and Chinese Chives

**DOI:** 10.3390/molecules25153391

**Published:** 2020-07-27

**Authors:** Yanjie Li, Quanshun An, Changpeng Zhang, Canping Pan, Zhiheng Zhang

**Affiliations:** 1Institute of Quality and Standard for Agro-Products, Zhejiang Academy of Agricultural Sciences, Hangzhou 310021, China; gqblyj@163.com (Y.L.); cpzhang1215@126.com (C.Z.); 2Department of Applied Chemistry, College of Science, China Agricultural University, Beijing 100193, China; anquanshun@cau.edu.cn

**Keywords:** Sin-QuEChERS nano, d-SPE, matrix effect, pesticide residue

## Abstract

In this study, a new rapid cleanup method was developed for the analysis of 111 pesticide multi-residues in lettuce and Chinese chives by GC–MS/MS and LC–MS/MS. QuEChERS (quick, easy, cheap, effective, rugged and safe)-based sample extraction was used to obtain the extracts, and the cleanup procedure was carried out using a Sin-QuEChERS nano cartridge. Comparison of the cleanup effects, limits of quantification and limits of detection, recoveries, precision and matrix effects (MEs) between the Sin-QuEChERS nano method and the classical dispersive solid phase extraction (d-SPE) method were performed. When spiked at 10 and 100 μg/kg, the number of pesticides with recoveries between 90% to 110% and relative standard deviations < 15% were greater when using the Sin-QuEChERS nano method. The MEs of Sin-QuEChERS nano and d-SPE methods ranged between 0.72 to 3.41 and 0.63 to 3.56, respectively. The results verified that the Sin-QuEChERS nano method was significantly more effective at removing pigments and more convenient than the d-SPE method. The developed method with the Sin-QuEChERS nano cleanup procedure was applied successfully to determine pesticide residues in market samples.

## 1. Introduction

Pesticides prevent and control pests, diseases and weeds and are used to improve vegetable production levels and quality. However, the application of pesticides often leaves pesticide residues in vegetables, which represent a health risk to humans and animals. Measuring trace levels of pesticides in vegetables is becoming a challenging task because of the increasing number and variety of pesticides, and the presence of sample matrix components.

In the past few decades, many extraction and cleanup methods for removing pesticide residues in vegetables have been developed, such as matrix solid-phase dispersion [1,2], solid phase extraction [3,4] and QuEChERS (quick, easy, cheap, effective, rugged and safe) [5]. The QuEChERS method consists of three main steps: extraction with organic solvent, partition with salts and cleanup by dispersive solid phase extraction (d-SPE) with a small quantity of sorbents. In the d-SPE procedure, primary secondary amine (PSA), graphitized carbon black (GCB) and octadecyl-silane (C_18_) are the main sorbents used to adsorb organic acids, and to some extent adsorb various pigments and sugars [5].

Since development, many government and organizations have used the QuEChERS method extensively because this is the most efficient method to remove pesticide multi-residues, especially from vegetables, fruits and many other matrices [6,7,8,9,10]. Nonetheless, development of more efficient and reliable methods for determining multi-class and multi-residue pesticides in vegetables continues. The QuEChERS method has been modified to increase its efficiency and effectiveness, especially the sorbents of d-SPE and cleanup methods. Multi-walled carbon nanotubes (MWCNTs) were used as alternative d-SPE sorbents in pesticide multi-residue analysis with the QuEChERS method [11,12,13,14]. Many advanced cleanup techniques were developed or modified, such as disposable pipette extraction [15], multiplug filtration cleanup (m-PFC) [16] and automated m-PFC [17,18]. The Sin-QuEChERS nano cleanup method is a newly developed cleanup technique, which has been used for detecting pesticide residues in tea [19], wolfberry [20], pepper and chili pepper [21] and other matrices [22,23]. In the Sin-QuEChERS nano cleanup method, MWCNTs mixed with PSA, anhydrous magnesium sulfate (MgSO_4_) and other sorbents are packed into a cartridge. The Sin-QuEChERS nano cartridge is inserted into a 50 mL centrifuge tube, and the extract is purified as the cartridge is pressed downward. The Sin-QuEChERS nano cleanup method is found to be very easy and rapid because it reduces the transfer and vortex steps in the cleanup procedure.

Among the different vegetables, lettuce (*Lactuca sativa* L.) and Chinese chives (*Allium tuberosum* Rottler ex Spreng.) are representatives of simple and complex vegetable matrices, respectively. Lettuce is a kind of widely consumed leafy vegetable; however, the presence of residual pesticides is a concern to consumers [24]. Chinese chives are a complex matrix in pesticide residue analysis because they are rich in sulfur-containing compounds and pigments that may cause matrix interference during mass spectrometry analysis [25,26]. In the current study, a method with QuEChERS-based extraction and the Sin-QuEChERS nano cleanup was established and validated for determining 111 pesticides in two representative vegetables. The 111 pesticides were selected based on the registration and the routine monitoring in lettuce and Chinese chives by the Chinese government. Tandem mass spectrometry (MS/MS) combined with gas chromatography (GC) and liquid chromatography (LC) was used for qualitative and quantitative analysis of the 111 pesticides. In addition, the performance, cleanup effects and matrix effect of the Sin-QuEChERS nano and classical d-SPE methods were compared. We also aimed to simplify the pretreatment procedure, improve the efficiency of the method and extend the application of the Sin-QuEChERS nano cleanup method.

## 2. Results and Discussion

### 2.1. Optimization of Sin-QuEChERS nano Cleanup Procedures

The schematic diagram of the Sin-QuEChERS nano cleanup procedure is presented in Figure 1. PSA, MWCNTs, C_18_ and GCB are common adsorbents used to adsorb impurities in the cleanup procedure. Cartridge A packed with 90 mg PSA and 15 mg MWCNTs was used for the lettuce sample (simple matrix), and cartridge B packed with 15 mg MWCNTs, 90 mg PSA, 80 mg GCB and 80 mg C_18_ was used for the Chinese chives sample (complex matrix). Because the amount of absorbents is fixed, the volume of purified extract influences the purification effect and recovery directly. Thus, different volumes (1 to 8 mL) of extract were purified at a spiking level of 100 μg/kg. The results showed that recoveries of most analytes increased as the purified volume increased up to 4 mL and then gradually reached a steady level above a 4 mL loading. The recoveries of most target pesticides were in an acceptable range of 70%–120% when the purified volume reached 4 mL. At purified volumes greater than 4 mL, the color of the purified extract deepened as the volume increased above 4 mL. Therefore, the optimum volume of the purified extract was set at 4 mL for further examination.

### 2.2. Method Validation and Comparison of Sin-QuEChERS nano and d-SPE Cleanup Methods

#### 2.2.1. Linearity, Limits of Quantification (LOQs) and Limits of Detection (LODs)

Linearity was evaluated at five concentrations (10, 50, 100, 200 and 500 μg/L) by matrix-matched standard calibration. Good linearity was achieved for most pesticides with determination coefficients (*R*^2^) greater than 0.99, except for parathion-methyl and methomy.

The limits of quantification (LOQs) and limits of detection (LODs) are presented in Appendix A
Appendix A. For the sin-QuEChERS nano method, the LOQs and LODs of the 111 pesticides present in lettuce and Chinese chives were in the range of 0.3–10 μg/kg and 0.1–3.0 μg/kg, respectively. For the d-SPE method, the LOQs and LODs of the 111 pesticides present in lettuce and Chinese chives were in the range of 0.4–10 μg/kg and 0.1–3.0 μg/kg, respectively. Thus, there were negligible differences between the two cleanup methods based on the LOQs and LODs values.

#### 2.2.2. Recovery and Precision

Recovery and repeatability experiments were performed at two levels (10 and 100 μg/kg) with three replicates at each level to evaluate the accuracy and precision of the methods. Appendix A
Appendix A shows the results of average recoveries and relative standard deviations (RSDs) of pesticides. The recoveries were calculated using the matrix-matched standard calibration curves, according to the external standard method. In general, the recoveries of most pesticides for the two methods met the European (EU) Commission’s guidance requirements. For the sin-QuEChERS nano method, the recoveries of most pesticides were in the range of 73% to 136% (75%–136% for lettuce and 73%–119% for Chinese chives) except for cyprodinil, which was in the range of 25%–45%. This result was in agreement with Zou et al. [25] and Qin et al. [27], who concluded that the cleanup procedures with MWCNTs were not suitable for pesticides containing a plane polycyclic structure. For the d-SPE method, the recoveries of analyzed pesticides were in the range of 70% to 132% (70%–128% for lettuce and 70%–132% for Chinese chives). As shown by the recoveries and RSDs, the sin-QuEChERS nano method performed slightly better than the d-SPE method. A larger number of pesticides with recoveries between 90% to 110% and RSDs < 15% was achieved when using the sin-QuEChERS nano method.

#### 2.2.3. Cleanup Effect

As shown in Figure 2, the extracts of lettuce and Chinese chives samples purified by the Sin-QuEChERS nano cleanup method were almost colorless. The Sin-QuEChERS nano method displayed better performance in removing pigments than the d-SPE method, because of the excellent ability of the MWCNTs to adsorb interference substances. Figure 3 shows total ion chromatograms (TIC) of GC–MS/MS and LC–MS/MS for the blank lettuce using the two cleanup methods. The results demonstrate that the Sin-QuEChERS nano cleanup method performed better than d-SPE method with fewer and lower levels of chromatographic interference substances.

#### 2.2.4. Matrix Effect

Matrix effects (MEs) are very common in both GC–MS/MS and LC–MS/MS methods and should be assessed at the method validation stage. Because of the co-extracted analytes, MEs are regarded as a signal enhancement (ME > 1) or suppression (ME < 1) of the analyte. MEs were estimated via the ratio of the calibration curve slopes of matrix to solvent. Studies recommend that MEs can be ignored when the ME values are in the range of 0.9 to 1.1 [11]. If the ME cannot be ignored, using a matrix-matched standard is the most effective way to compensate for MEs.

The MEs in this study are listed in Appendix A
Appendix A. The MEs of the Sin-QuEChERS nano method were in the range of 0.72–3.41 (0.88–2.05 for lettuce and 0.72–34 for Chinese chives), and the MEs of the d-SPE method ranged between 0.63 and 3.56 (0.80–2.68 for lettuce and 0.63–3.56 for Chinese chives). Figure 4 shows that the MEs were more obvious in the GC–MS/MS analysis and primarily acted as a signal enhancement. The MEs are also matrix dependent. The MEs of Chinese chives are more robust when compared with that of lettuce. The Sin-QuEChERS nano method reduced the matrix effect more efficiently than the d-SPE method because of the better cleanup capacity.

### 2.3. Application to the Monitoring of Real Samples

The development of the QuEChERS-based extraction method with the Sin-QuEChERS nano cleanup method was used in real sample analysis. Fourteen Chinese chives samples and eighteen lettuce samples were purchased from supermarkets, farmer’s markets or local vegetable-production factories in Beijing. All samples were extracted and cleaned up according to Section 3.5 and Section 3.6. Pesticide residues encountered in the analyzed samples are shown in Table 1. Fifteen pesticide residues were detected in 12 lettuce samples in the range of 0.010–0.87 mg/kg. Nineteen pesticide residues were detected in 12 Chinese chives samples with the residues in the range of 0.01–1.5 mg/kg. Among all the detected pesticides, dimethomorph and clothianidin had the highest detection rate at 44.4% and 64.3% in lettuce and Chinese chives, respectively.

Table 1 list the maximum residue limits (MRLs) of the detected pesticides established by China [28], the Codex Alimentarius Commission (CAC) [29] and the EU [30]. Compared to the lowest MRLs, the residues of 5 pesticides in lettuce and 13 pesticides in Chinese chives were above the MRLs. The residues of dimethomorph in lettuce with a detection frequency of 44.4% were lower the lowest MRL (9 mg/kg) established by the CAC. The detection frequency of difenoconazole and dimethomorph in Chinese chives was up to 50% and 35.7%; however, China, CAC and EU have not set the MRLs. From this test, the developed method was confirmed to be suitable for determination of pesticide multi-residues in a variety of vegetables.

## 3. Materials and Methods 

### 3.1. Chemicals and Materials

Pesticide reference standards (purity > 95%) in this study were provided by the China Agricultural University (Beijing, China).

Methanol, acetonitrile and HPLC grade acetone were purchased from Fisher Chemicals (Fair Lawn, NJ, USA). Analytical grade formic acid (88%), NaCl (99.5%) and MgSO_4_ (98%) were purchased from Sino-pharm Chemical Reagent (Beijing, China). PSA was purchased from Agilent Technologies (Palo Alto, CA, USA).

Two types of Sin-QuEChERS nano cartridges (Figure 1) with different proportions of sodium sulfate (NaSO_4_), MgSO_4_, C18, PSA, MWCNTs and GCB were provided by Lumiere Technologies (Beijing, China). Sin-QuEChERS nano cartridge A: 15 mg MWCNTs + 90 mg PSA + 2 g NaSO_4_ + 0.6 g MgSO_4_; Sin-QuEChERS nano cartridge B: 15 mg MWCNTs + 90 mg PSA + 80 mg C_18_ + 80 mg GCB + 2 g NaSO_4_ + 0.6 g MgSO_4_.

### 3.2. Stock Solutions and Standards

The individual pesticide standard stock solutions were prepared by accurately weighing 5–50 mg of each pesticide standard in volumetric flasks and dissolving them each in 10 mL methanol, acetonitrile or acetone, depending on pesticide solubility. A composite working standard solution of 5 mg/L was prepared by combining aliquots of each pesticide standard stock solution and diluting them in acetonitrile. Three series of calibration standards with five concentrations (10, 50, 100, 200, 500 μg/L) were diluted in acetonitrile, lettuce and Chinese chives extracts. The stock standard solutions and working standard solution were stored at −20 °C.

One hundred and eleven pesticides were analyzed, among which 75 were analyzed by GC–MS/MS, 60 pesticides were analyzed by LC–MS/MS and 24 were analyzed by GC–MS/MS and LC–MS/MS.

### 3.3. GC–MS/MS Analytical Conditions

A Thermo Scientific TSQ Quantum XLS triple quadrupole mass spectrometer coupled with a Thermo Scientific Trace 1300 GC (San Jose, CA, USA) was used for GC–MS/MS analysis. An Rxi^®^-5SiL MS column (20 m × 0.18 mm I.D., 0.18 μm film thickness) from Restek Corporation (Bellefonte, PA, USA) was used for the chromatographic separation of the compounds. The column temperature program started from 40 °C (hold 0.6 min), increased to 180 °C at the rate of 30 °C/min, then increased to 280 °C at the rate of 10 °C/min, then increased to 290 °C at the rate of 20 °C/min, and held at this final temperature for 5 min. The temperature of the injector port was 250 °C, and a 1 μL volume was injected into the splitless mode with a split flow of 50 mL/min and a splitless time of 1.0 min. The helium carrier gas flow rate was 0.85 mL/min. The ion source and transfer line temperature were 280 °C. The MS was operated in the electron ionization mode at 70 eV with the selective reaction monitoring (SRM) data acquisition mode. The MS/MS parameters are shown in Appendix A
Appendix A.

### 3.4. LC–MS/MS Analytical Conditions

The LC system was a Dionex UltiMate 3000 liquid chromatograph system (Dionex, CA, USA) with a quaternary pump (HPG-3400RS), autosampler (WPS-3000) and column oven (TCC-3000) equipped with a reversed-phase Syncronis C_18_ column (100 mm × 2.1 mm I.D., 1.7 μm particle size) from Thermo Scientific (San Jose, CA, USA). The column temperature was maintained at 40 °C, and the injection volume was 5 μL. Mobile phase A was water with 0.1% formic acid, and mobile phase B was acetonitrile. The separation was performed at a flow rate of 0.4 mL/min, and the gradient elution was 0–18 min, linear gradient 20%–100% B, held for 2 min; 20–21 min, linear gradient 100%–20% B, held for 2 min.

The LC system was connected to a Thermo Scientific TSQ Quantum Access MAS triple stage MS/MS (San Jose, CA, USA) equipped with an electrospray ionization (ESI) source in the positive and negative ion mode. The SRM mode was used, and the *m*/*z* ratios of all target pesticides are listed in Appendix A
Appendix A, along with the tube lens voltage and the collision energy. The capillary temperature, vaporizer temperature, aux gas (N_2_) pressure and sheath gas (N_2_) pressure were set to 300 °C, 300 °C, 15 Arb and 35 Arb, respectively. The spray voltage for positive and negative polarity was set to 3500 V and 2300 V, respectively.

### 3.5. Sample Preparation

The lettuce and Chinese chives samples were homogenized with a blender for 30 s. The homogenized lettuce and Chinese chives samples were then extracted as described by Lehotay [5] with some modifications. Briefly, the sample was weighed into a 50 mL Teflon centrifuge tube. After adding 4 g anhydrous MgSO_4_ and 1 g NaCl and shaking for 1 min, the tube was cooled in an ice-water bath for 5 min and then centrifuged for 5 min at 3800 rpm. The supernatant was used for further cleanup.

For recovery experiments, the homogenized blank samples (10 ± 0.1 g) were spiked by the addition of the working standard solution at two concentration levels of 10 and 100 μg/kg and left for at least 30 min before extraction.

### 3.6. Sin-QuEChERS nano and d-SPE Cleanup Procedures

For the Sin-QuEChERS nano cleanup procedure, the schematic of the steps used is presented in Figure 1. The Sin-QuEChERS nano cartridge was inserted into the 50 mL Teflon centrifuge tube and pressed down at a steady speed of ~1 mm/s until 4 mL purified extract was achieved. Finally, 1 mL purified extract was filtered through a 0.22-μm membrane and placed into an auto-sampler vial for GC–MS/MS or LC–MS/MS analysis.

For the dispersive-SPE cleanup procedure, 1 mL extract was transferred into a 2-mL microcentrifuge vial containing 50 mg PSA and 150 mg MgSO_4_. The mixture was eddied with a vortex mixer for 30 s and centrifuged for 1 min at 10,000 rpm. Finally, the purified extract was filtered through a 0.22-μm membrane and placed into an auto-sampler vial for GC–MS/MS or LC–MS/MS analysis.

### 3.7. Method Validation

The methods were validated according to the EC guidance document SANTE/11813/2017 [31]. Analytical parameters evaluated were linearity, accuracy and precision, LOD, LOQ and ME. The LOD and LOQ were calculated by the lowest concentration that produced signal-to-noise ratios of 3 and 10, respectively. Matrix effects were estimated by comparing the calibration curves slopes of matrix and solvent.

### 3.8. Confirmation Criteria

The MS coupled GC or LC separation system can provide the retention time (Rt), *m*/*z* ratio and relative abundance data simultaneously. For identification of the analytes in the extract, the Rt and relative ion ratio criterion were used. According to the EC guidance document [31], the Rt of the analyte from the sample extract should correspond to the calibration standard with a tolerance of ±0.1 min, and the relative ion ratio from sample extracts should be within ± 30% of the average of the calibration standard.

## 4. Conclusions

In this study, a rapid and sensitive method for the analysis of pesticide multi-residues in lettuce and Chinese chives was developed using the Sin-QuEChERS nano cleanup method combined with the QuEChERS-based sample extraction method. The method was validated through linearity, LODs and LOQs, accuracy and precision and matrix effect. The recoveries of 110 pesticides were in the range of 73% to 136% (75%–136% for lettuce and 73%–119% for Chinese chives) except for cyprodinil, which was in the range of 25%–45%. The LODs and LOQs for the 111 pesticides were in the range of 0.3–10 μg/kg and 0.1–3.0 μg/kg, respectively. Fourteen Chinese chives samples and eighteen lettuce samples were analyzed by the developed method. The results indicated that the proposed method to determine various classes of pesticide residues is sensitive.

Meanwhile, comparison of the cleanup effects, limits of quantification and limits of detection, recoveries, precision and MEs between the Sin-QuEChERS nano method and the classical d-SPE method were performed; comparison of the cleanup method, detection limit, recovery, advantages and disadvantages between the Sin-QuEChERS nano method reported in this study and the published methods were made (as shown in Table 2). The results verified that the Sin-QuEChERS nano method performed better with respect to the cleanup effect, especially for the removal of pigments, and is a simplified and efficient extract purification procedure.

## Figures and Tables

**Figure 1 molecules-25-03391-f001:**
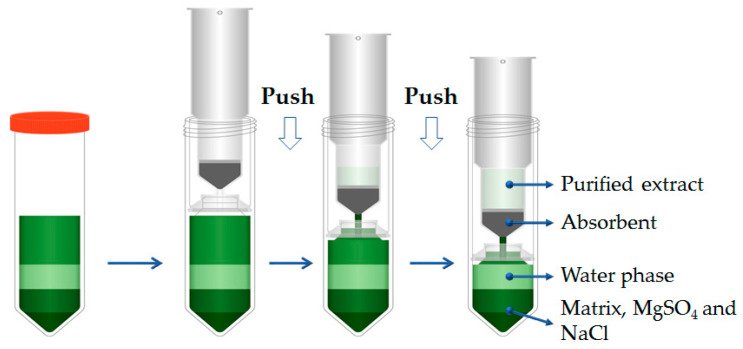
Schematic of the Sin-QuEChERS nano cleanup method.

**Figure 2 molecules-25-03391-f002:**
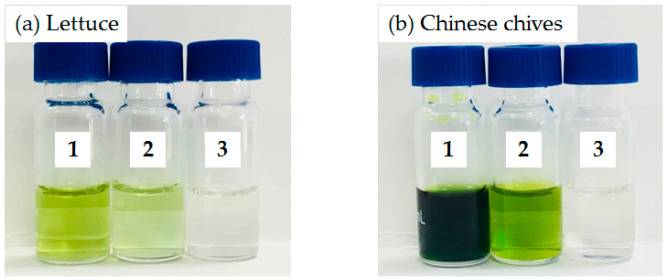
Comparison of the pigment-cleanup effects between Sin-QuEChERS nano and d-SPE methods. (**a**) Extract of lettuce (1) without cleanup, (2) cleaned-up by d-SPE and (3) cleaned-up by Sin-QuEChERS nano. (**b**) Extract of Chinese chives (1) without cleanup, (2) cleaned-up by d-SPE and (3) cleaned-up by Sin-QuEChERS nano.

**Figure 3 molecules-25-03391-f003:**
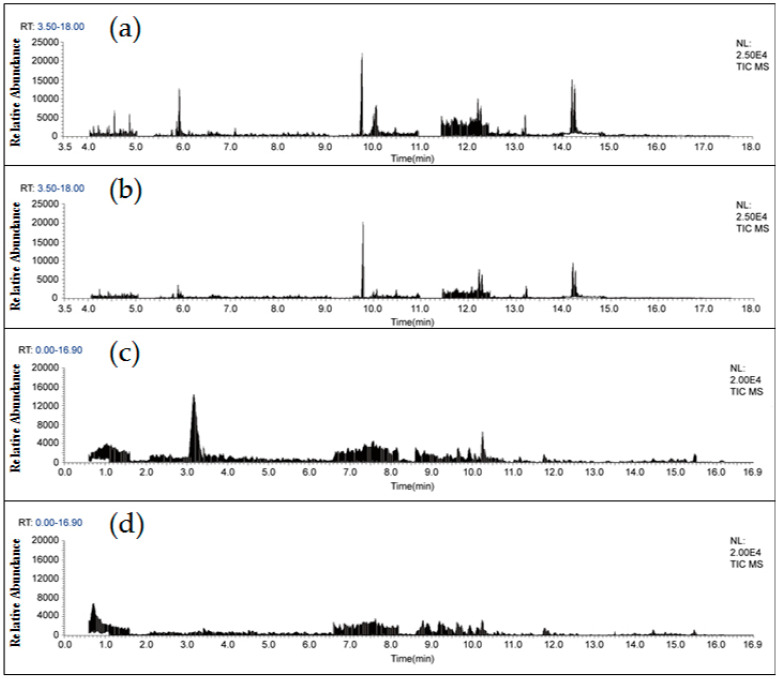
Total ion chromatograms (TIC) for a blank lettuce extract with different cleanup procedures. (**a**) TIC of GC–MS/MS with d-SPE cleanup, (**b**) TIC of GC–MS/MS with Sin-QuEChERS nano cleanup, (**c**) TIC of LC–MS/MS with d-SPE cleanup and (**d**) TIC of LC–MS/MS with Sin-QuEChERS nano cleanup.

**Figure 4 molecules-25-03391-f004:**
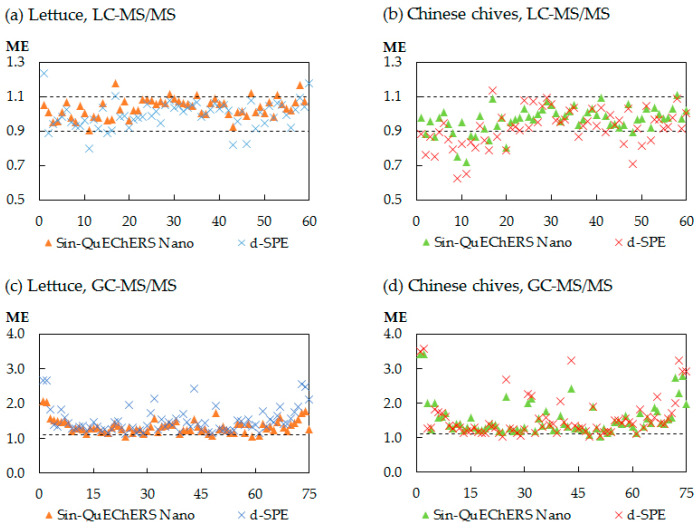
Comparison of the matrix effects (MEs) for all pesticides in lettuce and Chinese chives using the different cleanup procedures. (**a**) pesticides in lettuce analyzed by LC–MS/MS, (**b**) pesticides in Chinese chives analyzed by LC–MS/MS, (**c**) pesticides in lettuce analyzed by GC–MS/MS and (**d**) pesticides in Chinese chives analyzed by GC–MS/MS.

**Table 1 molecules-25-03391-t001:** Pesticide residues in real lettuce and Chinese chives samples.

Vegetable	Pesticide	Positive Samples		Samples Exceed MRL		Range of Residues (mg/kg)	LOQ (μg/kg)	MRL (China/CAC/EU) (mg/kg)
		N	%	N	%			
Lettuce	Difenoconazole	3	16.7	-	-	0.041–0.87	1.2	2/2/4
	Imidacloprid	1	5.6	-	-	0.29	3.7	1/-/2
	Propiconazole	4	22.2	3 ^EU^	16.7	0.010–0.30	1.2	-/-/0.01 *
	Hexaconazole	1	5.6	1 ^EU^	5.6	0.013	2.3	-/-/0.01 *
	Myclobutanil	1	5.6	-	-	0.020	0.3	0.05 ^a^/0.05 ^a^/0.05
	Cyhalothrin	2	11.1	-	-	0.017–0.062	5.7	2/-/0.15
	Cypermethrin	1	5.6	-	-	0.056	1.6	2/0.7 ^a^/2
	Clothianidin	1	5.6	-	-	0.015	8.4	2 ^a^/2 ^a^/0.1
	Thiametoxam	1	5.6	-	-	0.19	1.5	3 ^a^/-/-
	Thifluzamide	1	5.6	-	-	0.015	3.3	-/-/-
	Hexythiazox	1	5.6	-	-	0.026	1.0	-/-/0.5
	Buprofezin	2	11.1	2 ^EU^	11.1	0.012–0.030	1.9	-/-/0.01 *
	Triadimefon	1	5.6	1 ^EU^	5.6	0.035	1.0	-/-/0.01 *
	Cymoxanil	1	5.6	1 ^EU^	5.6	0.12	7.4	-/-/0.03
	Dimethomorph	8	44.4	-	-	0.011–0.32	4.1	-/9/15
Chinese chives	Difenoconazole	7	50.0	-	-	0.010–0.11	1.2	-/-/-
	Pyridaben	2	14.3	1 ^EU^	7.1	0.010–0.011	0.7	-/-/0.01 *
	Chlorpyrifos	4	28.6	2 ^China^	14.3	0.011–1.4	10	0.1/-/-
	Bifenthrin	1	7.1	1 ^EU^	7.1	0.031	0.9	- ^c^/-/0.01 *
	Cyhalothrin	5	35.7	2 ^CAC^	14.3	0.017–0.50	6.0	0.5 ^c^/0.2 ^b^/0.2
	Cypermethrin	4	28.6	1 ^China^	7.1	0.054–1.5	7.7	1 ^c^/-/-
	Kresoxim-methyl	1	7.1	-	-	0.38	2.9	-/-/-
	Azoxystrobin	2	14.3	-	-	0.012–0.021	0.8	1 ^b^/10 ^b^/10 ^b^
	Esfenvalerate	1	7.1	1 ^EU^	7.1	0.020	7.0	-/-/0.02 *
	Clothianidin	9	64.3	9 ^EU^	64.3	0.033–0.38	3.9	- ^c^/-/0.01 *
	Thifluzamide	1	7.1	-	-	0.035	3.3	-/-/-
	Triadimenol	1	7.1	1 ^EU^	7.1	0.018	2.2	-/-/0.01 *
	Triazophos	1	7.1	1 ^EU^	7.1	0.013	1.5	-/-/0.01 *
	Triadimefon	1	7.1	1 ^EU^	7.1	0.017	5.0	-/-/0.01 *
	Trifloxystrobin	1	7.1	-	-	0.41	1.7	0.7/-/-
	Dimethomorph	5	35.7	-	-	0.010–0.38	3.9	-/-/-
	Phosmet	1	7.1	1 ^EU^	7.1	0.41	2.1	-/-/0.05 *
	Omethoate	1	7.1	1 ^China^	7.1	0.32	3.1	0.02 ^b^/-/-
	Diethofencarb	1	7.1	1 ^EU^	7.1	0.019	2.8	-/-/0.01 *

^a^ Maximum residue limit (MRL) of the group of leafy vegetables; ^b^ MRL of the group of bulb vegetables; ^c^ registered in China on this crop; * indicates lower limit of analytical determination.

**Table 2 molecules-25-03391-t002:** Comparison of the quality parameters for determining pesticide residues in lettuce and Chinese chives between the developed method and reported methods.

Reference	Samples	Pesticides	Extraction Method	Cleanup Method	Detection	Recovery	Detection Limits	Advantages	Disadvantages
Zhu et al. [32]	Garlic chives	51	Acetonitrile	d-SPE (GCB, PSA and MgSO_4_)	GC–MS/MS	60.4%–111.5% (20.34%–36.7% for amitraz)	0.8–25 μg/kg (LOD)	Economic; easy.	High detection limits; shaky purification and centrifugal separation.
Ying et al. [33]	Chinese chives	13	Acetonitrile	none	GC–FPD	78%–115%	0.01–0.03 mg/kg (LOD)	Low operation fee; without centrifuge steps.	Reagents and time consuming; matrix effects; high detection limits.
Han et al. [34]	Leaf lettuce	70	Acetonitrile	d-SPE (MWCNTs and MgSO4)	LC–MS/MS	74%–119%	0.3–6.2 μg/kg (LOQ)	Good cleanup performance; high sensitivity.	Shaky purification and centrifugal separation.
Ribeiro Begnini Konatu et al. [35]	Lettuce	16	Acetonitrile, citrate buffer	d-SPE (GCB, PSA and MgSO_4_)	LC–MS/MS	79%–115%	5–3200 μg/kg (LOQ)	Economic; quick.	High detection limits; sorbents consuming,
This study	Lettuce	111	Acetonitrile	Sin-QuEChERS nano cleanup method (MWCNTs, PSA, C18 and MgSO4)	GC–MS/MS and LC–MS/MS	75%–136% (41%–45% for cyprodinil)	0.3–10 μg/kg (LOQ)	Purification and separation in one step; easier and quicker.	Excess purified extracts.
	Leaf lettuce					73%–119% (25%–28% for cyprodinil)	0.4–10 μg/kg (LOQ)		

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
