# Peer review of "Comparison of Sin-QuEChERS Nano and d-SPE Methods for Pesticide Multi-Residues in Lettuce and Chinese Chives"

_molecules, 2020, doi:10.3390/molecules25153391_

Round 1

Reviewer 1 Report

Comments and remarks:

  • the abstract part should be re-written, it is too chaotic, the English language should be check, there is no details about highlights,
  • the title is too long, please change it,
  • the introduction part should be re-written, the english should be check,
  • in the introduction part should be highlighted the main aim of the paper, and additionally, what is the novelty of carried research work,
  • the provided highlights are not sufficient, please provide a new one,
  • the list of abbreviation should be added,
  • in the Reference list there is lack of the most important papers regarding biological activity, correlation between the in silico, in vitro and in vivo studies, sample preaparation techniques in case of pesticides,
  • how do the Authors select the analytes? The rational of the choice of the selected biologically active compounds studied is missing and should be clearly discussed. Additionally, these analytes are not listed in the abstract section,
  • the main results achieved should be indicated, I mean the Authors should target to the points. This rather appears as a summary of what has been done,
  • what about validation of the obtained models (training and test sets)?,
  • what is a value of mean square errors (MSE)?,
  • it would be interesting to see data (e.g. chromatograms) from real biological samples but not spiked,
  • did the Authors performed any optimization stages in the case of sample preparation?
  • Authors should present their data in the table in comparison to the existed methods in the literature.

Reviewer 2 Report

I suggest move the Table 1 to supplementary material. It is so long comparing with the procedures in the original manuscript.

I recommend include a new Table with the limit and averange of the quality parameter of the presented methods (both, sin-QuEChERS and d-SPE)

The clean up procedure was based in previous studies? Were the amount of sorbents and solids optimized? Please refer it in the manuscript.

Round 2

Reviewer 1 Report

-